

# The use of chloroplast genome sequences to solve phylogenetic incongruences in *Polystachya* Hook (Orchidaceae Juss)

Narjara Lopes de Abreu[1,2], Ruy José Válka Alves[2], Sérgio Ricardo Sodré Cardoso[3], Yann J.K. Bertrand[1], Filipe Sousa[1,4], Climbiê Ferreira Hall[5], Bernard E. Pfeil[1] and Alexandre Antonelli[1,6,7,8]

[1] Department of Biological and Environmental Sciences, University of Gothenburg, Göteborg, Sweden
[2] Museu Nacional, Universidade Federal do Rio de Janeiro, São Cristóvão, Rio de Janeiro, Brasil
[3] Instituto de Pesquisas, Jardim Botânico do Rio de Janeiro, Diretoria de Pesquisa Científica, Rio de Janeiro, Brasil
[4] Centro de Ciências do Mar, Universidade do Algarve, Faro, Portugal
[5] Campus Três Lagoas, Universidade Federal de Mato Grosso do Sul, Três Lagoas, Mato Grosso do Sul, Brasil
[6] Gothenburg Botanical Garden, Göteborg, Sweden
[7] Gothenburg Global Biodiversity Centre, Göteborg, Sweden
[8] Department of Organismic and Evolutionary Biology, Harvard University, Cambridge, MA, USA

Corresponding author
Narjara Lopes de Abreu,
narjara.lopes@gmail.com

## ABSTRACT

**Background:** Current evidence suggests that for more robust estimates of species tree and divergence times, several unlinked genes are required. However, most phylogenetic trees for non-model organisms are based on single sequences or just a few regions, using traditional sequencing methods. Techniques for massive parallel sequencing or next generation sequencing (NGS) are an alternative to traditional methods that allow access to hundreds of DNA regions. Here we use this approach to resolve the phylogenetic incongruence found in *Polystachya* Hook. (Orchidaceae), a genus that stands out due to several interesting aspects, including cytological (polyploid and diploid species), evolutionary (reticulate evolution) and biogeographical (species widely distributed in the tropics and high endemism in Brazil). The genus has a notoriously complicated taxonomy, with several sections that are widely used but probably not monophyletic.

**Methods:** We generated the complete plastid genome of 40 individuals from one clade within the genus. The method consisted in construction of genomic libraries, hybridization to RNA probes designed from available sequences of a related species, and subsequent sequencing of the product. We also tested how well a smaller sample of the plastid genome would perform in phylogenetic inference in two ways: by duplicating a fast region and analyzing multiple copies of this dataset, and by sampling without replacement from all non-coding regions in our alignment. We further examined the phylogenetic implications of non-coding sequences that appear to have undergone hairpin inversions (reverse complemented sequences associated with small loops).

**Results:** We retrieved 131,214 bp, including coding and non-coding regions of the plastid genome. The phylogeny was able to fully resolve the relationships among all species in the targeted clade with high support values. The first divergent

species are represented by African accessions and the most recent ones are among Neotropical species.

**Discussion:** Our results indicate that using the entire plastid genome is a better option than screening highly variable markers, especially when the expected tree is likely to contain many short branches. The phylogeny inferred is consistent with the proposed origin of the genus, showing a probable origin in Africa, with later dispersal into the Neotropics, as evidenced by a clade containing all Neotropical individuals. The multiple positions of *Polystachya concreta* (Jacq.) Garay & Sweet in the phylogeny are explained by allotetraploidy. *Polystachya estrellensis* Rchb.f. can be considered a genetically distinct species from *P. concreta* and *P. foliosa* (Lindl.) Rchb.f., but the delimitation of *P. concreta* remains uncertain. Our study shows that NGS provides a powerful tool for inferring relationships at low taxonomic levels, even in taxonomically challenging groups with short branches and intricate morphology.

# INTRODUCTION

Orchidaceae is considered the largest family of flowering plants, with over 25,000 species (*Dressler, 1990*; *Christenhusz & Byng, 2016*). The family probably dates back to the Late Cretaceous, as indicated by fossil-calibrated molecular phylogenies (*Gustafsson, Verola & Antonelli, 2010*; *Ramírez et al., 2007*, *2011*). *Polystachya* Hook. is an orchid genus containing 240 species, with most species found in Africa (*Dressler, 1993*). A total of 13 species are reported from the Neotropical region (*Mytnik-Ejsmont, 2011*), but this number may increase when considering the endemic species from Brazil that were not accounted for in *Mytnik-Ejsmont (2011)* or were considered synonymous (*Barros et al., 2010*).

Recent studies have shown a number of peculiar cytological, evolutionary and biogeographic aspects of *Polystachya*. The genus has diploid and polyploid species; the latter recently formed in the Neotropics and Madagascar (*Rupp et al., 2010*; *Russell et al., 2010b*). Unlike most genera of Orchidaceae, *Polystachya* has a wide geographical distribution range (*Pridgeon et al., 2005*; Fig. 1), having species that are Pantropical or have a transatlantic distribution. On the other hand, the Neotropics presents a high level of endemism. Brazil, as an example, has 12 species of which 10 are endemic (*Barros et al., 2010*). In addition, there is evidence of reticulate evolution in the genus and hybridization with independent origins (*Russell et al., 2010a*).

The monophyly of the genus has been reported in the latest studies (*Russell et al., 2010b*; *Mytnik-Ejsmont, 2011*), which contrasts starkly with the low level of monophyly observed in the taxonomic sections described within the genus. Those 15 sections (*Kraenzlin, 1926*; *Summerhayes, 1942*, *1947 apud Russell et al., 2011*; *Brenan, 1954*; *Cribb, 1978*) are based on morphological characters and have been useful for field identification and inventories, but do not find support as natural groupings in the molecular studies currently available (*Russell et al., 2010b*; *Mytnik-Ejsmont, 2011*). According to those
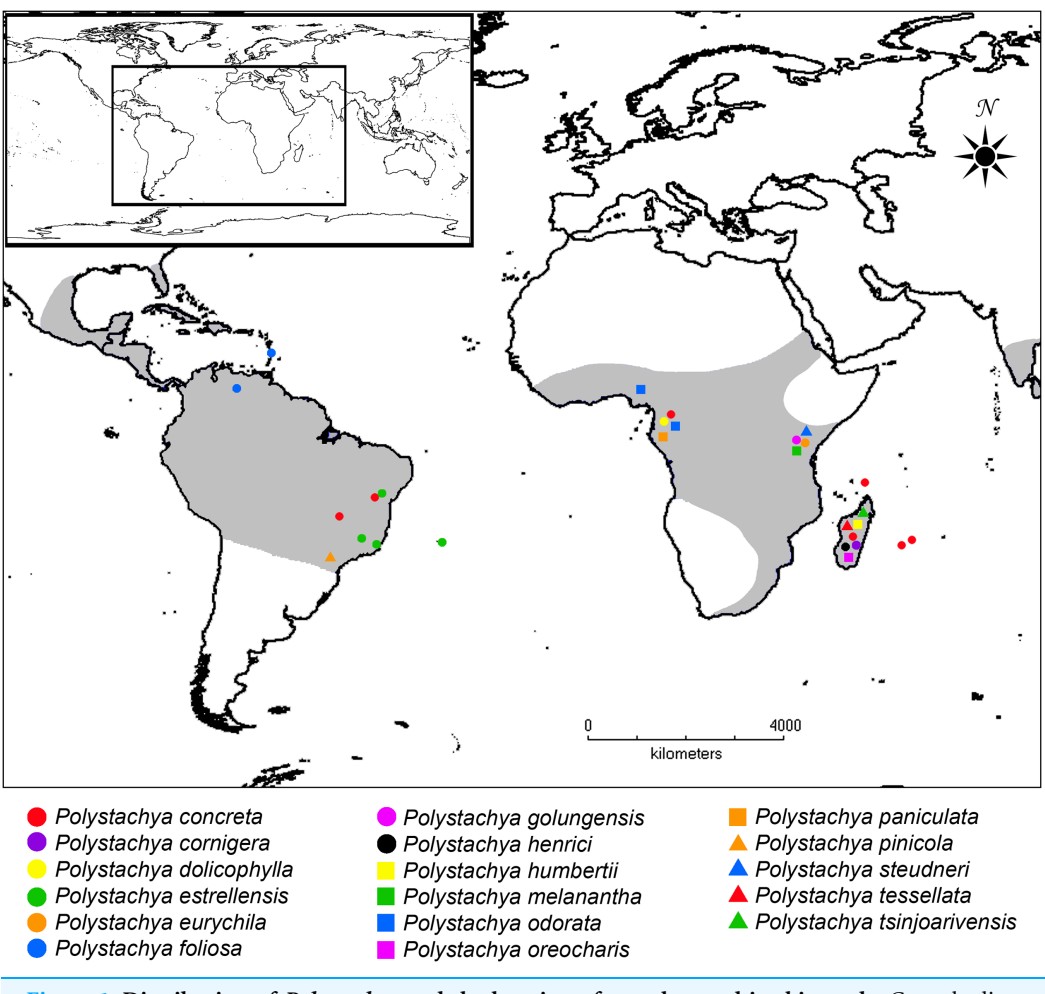

**Figure 1 Distribution of *Polystachya* and the location of samples used in this study.** Gray shading shows the distribution of the genus. Colored symbols show the location of samples used here, with the species determination of each sample as per Table 1. 

molecular studies, all sections are polyphyletic or paraphyletic, except sect. *Isochiloides* (*Russell et al., 2010b*).

Section *Polystachya* has been described as comprising 32 species worldwide and is the only section with species of pantropical distribution (*Mytnik-Ejsmont, 2011*). However, according to molecular analyses, some species of this section appear to be more related to species of other sections (*Russell et al., 2010b*; *Mytnik-Ejsmont, 2011*). These studies highlight the need for new infrageneric divisions based on robust molecular evidence. *Russell et al. (2010b)*, using chloroplast markers, defined five different clades that could be used as the basis for a revised classification of new sections within the genus. Clade III (sensu *Russell et al., 2010b*) includes species from five different sections (sects. *Polystachya*, *Eurychilae*, *Caulescentes*, *Superpositae*, *Polychaete*) and is divided into distinct subclades of morphologically diverse plants. These species are Pantropical (such as *P. concreta*), Neotropical (such as *P. foliosa*), Malagasy endemics (such as *P. henrici*) or African (such as *P. odorata*). The relationships among Clade III species remain unresolved because

specimens of *P. concreta*, *P. foliosa*, *P. henrici* and *P. modesta* form a large polytomy, due to low levels of divergence between sequences. In addition, several distinct subclades are unresolved at the base. The molecular phylogenetic studies produced so far have included about 35% of the recognized species within the genus, and used only a small number of nuclear (*PgiC* between exons 11 and 15, *PhyC* exon 1, *Rpb2* intron 23 and ITS in *Russell et al., 2010a*; only ITS in *Mytnik-Ejsmont, 2011*) and chloroplast markers (*rps16* intron, *rps16* exon 2, *rps16-trnK* spacer, *trnK* intron excluding *matK*, *matK* and *psbD-trnT* spacer in *Russell et al., 2010b*; *rps16-trnK*, *rps16* and *rpl32-trnL* in *Mytnik-Ejsmont, 2011*).

New methods of DNA sequencing as well as the development of more powerful algorithms are propelling the replacement of trees generated from one or a few genes to those constructed from hundreds of them (*Edwards, 2009*). The improvement of massively parallel sequencing techniques—or next generation sequencing (NGS)—has increased the amount of data available for biological research, whether the fully annotated reference genomes of species under study have been sequenced or not (*Bräutigam & Gowik, 2010*). However, despite its obvious potential, NGS technology is underused in most studies of plant systematics (*Cronn et al., 2012*; *Carstens et al., 2013*; *Eaton & Ree, 2013*), probably as a result of a prevailing focus on non-model organisms (which require de novo genomic sequencing and its inherent challenges), the need to sample many individuals per species and the absence of well-established protocols (*McCormack et al., 2013*). One method that increases the efficiency of NGS for non-model species compared to other genomic partitioning strategies is sequence capture (or hybridization-based enrichment), which is based on the prior selection of loci of interest (*Lemmon & Lemmon, 2013*). The main benefit of this technique is that the number of specific sequences obtained can be very high, which makes it an advantageous method compared to PCR-based approaches if the objective is to sequence several individuals and multiple loci. Furthermore, sequence capture when combined with NGS platforms, such as Illumina, also reduces the costs of the process (*Lemmon, Emme & Lemmon, 2012*). Sequence capture methods have successfully been used in other plant genera to generate large amounts of useful data for phylogenetic inference (*Kamneva et al., 2017*; *Sousa et al., 2014*; *Stephens et al., 2015*; *Weitemier et al., 2014*).

The necessity of a molecular phylogenetic framework for (and a morphological taxonomic revision of) *Polystachya* is clear. It requires a well-resolved phylogenetic hypothesis in order to clarify the relationships between species and also to redefine new infrageneric sections. In this paper, we explore the use of nearly complete plastid genomes (note that we use chloroplast and plastid interchangeably when referring to these genomes), obtained by sequence capture and massively parallel sequencing, to solve the phylogenetic inconsistencies found within Clade III of *Polystachya* (sensu *Russell et al., 2010b*). We also explore whether sequencing the entire chloroplast genome using NGS was worthwhile, compared to PCR and Sanger sequencing of a few fast-evolving loci. We hope that the results generated here can be extended to the rest of the genus and thus result in new interpretations of the evolutionary and biogeographic history of the group.

## MATERIALS AND METHODS

### Sampling and DNA extraction

We sampled 19 species and 48 individuals (Table 1; Fig. 1), of which 15 were collected in different locations in continental Brazil and three were collected on Trindade Island in the South Atlantic. The DNA of the 15 Brazilian samples was extracted from 10 mg of tissue dried with silica gel and using the DNeasy Plant Mini Kit (Qiagen, Valencia, CA, USA). DNA samples for the remaining 33 individuals were provided by the University of Vienna and the DNA bank of the Royal Botanic Gardens, Kew. To this sample of individuals, selected because they cluster in Clade III (as defined in *Russell et al., 2010b*), we added *Polystachya tessellata* Lindl., supposedly synonymous of *P. concreta*. We also included multiple samples of *P. concreta*, because previous studies have reported a lack of monophyly for this species and several synonymous species. Permits to collect were provided by the Ministério do Meio Ambiente (MMA), Instituto Chico Mendes de Conservação da Biodiversidade (ICMBio) and Sistema de Autorização e Informação em Biodiversidade (SISBIO), with registration number 29478-1.

*Polystachya bicolor* Rolfe and *P. melanantha* Schltr. were chosen as possible more distant relatives of the species in focus, in order to provide additional context for the phylogenetic inference. All studies conducted to date resolve *P. melanantha* as an outgroup with respect to the Clade III species. *Polystachya bicolor* has already been treated as a synonym of *P. rosea* (*Mytnik-Ejsmont, 2011*), with an uncertain position in the phylogenetic trees generated thus far, being sometimes closely related to *P. concreta* and other associated species (*Mytnik-Ejsmont, 2011*) and sometimes closely related to species from other clades (*Russell et al., 2010b*; *Mytnik-Ejsmont, 2011*).

### Probe design for DNA capture

We used complete chloroplast genome of *Phalaenopsis aphrodite* subsp. *formosana* (NC_007499.1) (*Chang et al., 2006*) as the reference for the design of capture probes, because there is no completely sequenced chloroplast genome of a *Polystachya*. According to molecular analyses, *Polystachya* and *Phalaenopsis* belong to different sub-tribes but are closely related within the Vandeae tribe (*Van Den Berg et al., 2005*; *Górniak, Paun & Chase, 2010*; *Freudenstein & Chase, 2015*). The use of a quite distantly related species is made possible by the DNA capture kit (MYcroarray, Ann Arbor, MI, USA), which is able to support differences larger than 5% between probe sequences and target sequences (*Li et al., 2013*). The complete sequence of the chloroplast genome of *Phalaenopsis aphrodite* subsp. *formosana* was divided into blocks of 360 bp. Every second block was used as the template for probe design; the probes consisting of 120 bp sequences, three to each block without overlap. Given the genomic DNA fragment sizes of between 300 and 400 bp (below) and that fragments can contain complementary sequence anywhere on their length to attach to a probe, fragments can contain up to 200–300 bp of genomic sequence into the flanking regions beyond the probes, or cover the probes with little extent into the flanking sequence, or somewhere in between. In this way, captured sequences produce a series of tiled overlapping sequence for high quality genomic assembly.

**Table 1 Species and sample information of the accessions used in the study.**

| Species | Location | Voucher | ENA reference | Before genomic library assembly | | | After genomic library assembly | |
|---|---|---|---|---|---|---|---|---|
| | | | | Concentration (ng/uL) | Purity 260/280 | Volume | Concentration (ng/uL) | Purity 260/280 |
| P. bicolor (=P. rosea) | – | Kew 25884 | ERS2203588 | 20.1 | 1.51 | 50 | 31.7 | 1.93 |
| P. concreta (1) | Brazil, Distrito Federal | N.L.Abreu 254 | ERS2203559 | 39.2 | 1.92 | 75 | 17.2 | 1.92 |
| P. concreta (2) | Brazil, Distrito Federal | N.L.Abreu 254 | ERS2203560 | 33.6 | 1.96 | 75 | 31.9 | 1.9 |
| P. concreta (3) | Brazil, Bahia State | N.L.Abreu 251 | ERS2203564 | 18 | 1.78 | 70 | 29.9 | 1.97 |
| P. concreta (4) | Brazil, Bahia State | N.L.Abreu 251 | ERS2203565 | 8.5 | 1.52 | 130 | 38.3 | 1.9 |
| P. concreta (5) | Cameroon | A. Russell 40 (YA) | ERS2203570 | 35 | 1.82 | 8 | 15 | 1.75 |
| P. concreta (6) | Brazil | HBV ORCH 066004 | ERS2203572 | 21.9 | 1.53 | 6 | 9.8 | 1.9 |
| P. concreta (7) | Mauritius | HBV ORCH 07278 | ERS2203575 | 26 | 1.78 | 13 | 32.4 | 1.85 |
| P. concreta (8) | Réunion | HBV "Chase & Samuel 1" | ERS2203576 | 34 | 1.82 | 13 | 30 | 1.85 |
| P. concreta (9) | Comoros | HBV ORCH 07417 | ERS2203577 | 23.7 | 1.83 | 13 | 30.4 | 1.86 |
| *P. concreta (10) | Madagascar | Fischer&Sieder FS3210 (WU) | – | 21 | 1.81 | 13 | 9.4 | 1.63 |
| *P. concreta (11) | Madagascar | Kew 17854 | – | 70 | 1.65 | 67 | 9.6 | 1.6 |
| *P. cornigera | Madagascar | Fischer&Sieder FS3208 (WU) | – | 22.2 | 1.76 | 11 | 7.7 | 1.58 |
| P. dolichophylla | Cameroon | Kew 25886 | ERS2203589 | 108.9 | 1.05 | 55 | 30.3 | 1.89 |
| P. estrellensis (1) | Brazil, Minas Gerais State | N.L.Abreu 255 | ERS2203551 | 29.8 | 1.81 | 65 | 33.7 | 1.9 |
| P. estrellensis (2) | Brazil, Minas Gerais State | N.L.Abreu 255 | ERS2203552 | 17.8 | 2.06 | 90 | 26 | 1.54 |
| P. estrellensis (3) | Brazil, Minas Gerais State | N.L.Abreu 255 | ERS2203553 | 11.9 | 1.82 | 75 | 35 | 1.86 |
| P. estrellensis (4) | Brazil, Espírito Santo State | N.L.Abreu 253 | ERS2203554 | 12.8 | 1.77 | 65 | 18.7 | 1.84 |
| P. estrellensis (5) | Brazil, Espírito Santo State | N.L.Abreu 253 | ERS2203555 | 22.4 | 1.87 | 70 | 21 | 1.96 |
| P. estrellensis (6) | Brazil, Espírito Santo State | N.L.Abreu 253 | ERS2203556 | 28.2 | 1.79 | 70 | 26.3 | 1.96 |
| P. estrellensis (7) | Brazil, Bahia State | N.L.Abreu 252 | ERS2203557 | 25.4 | 1.91 | 70 | 29.3 | 1.98 |
| P. estrellensis (8) | Brazil, Bahia State | N.L.Abreu 252 | ERS2203558 | 14.6 | 1.79 | 70 | 22.4 | 2.02 |
| P. estrellensis (9) | Brazil, São Paula State | N.L.Abreu 256 | ERS2203561 | 27.8 | 1.61 | 90 | 20.4 | 2.05 |
| P. estrellensis (10) | Brazil, São Paula State | N.L.Abreu 256 | ERS2203562 | 23.8 | 1.81 | 75 | 23.1 | 1.93 |
| P. estrellensis (11) | Brazil, São Paula State | N.L.Abreu 256 | ERS2203563 | 22.3 | 1.68 | 90 | 28 | 1.97 |

| Species | Location | Voucher | ENA reference | Before genomic library assembly | | | After genomic library assembly | |
|---|---|---|---|---|---|---|---|---|
| | | | | Concentration (ng/uL) | Purity 260/280 | Volume | Concentration (ng/uL) | Purity 260/280 |
| *P. estrellensis (12)* | *Brazil, Trindade's Island* | – | ERS2203566 | 10.1 | 1.64 | 130 | 30 | 1.88 |
| *P. estrellensis (13)* | *Brazil, Trindade's Island* | – | ERS2203567 | 12.5 | 1.64 | 130 | 30 | 1.83 |
| *P. estrellensis (14)* | *Brazil, Trindade's Island* | – | ERS2203568 | 10.3 | 1.48 | 130 | 34.7 | 1.84 |
| *P. eurychila* | *Kenya* | Kew 17963 | ERS2203586 | 207.6 | 0.94 | 85 | 13.7 | 1.98 |
| *P. foliosa (1)* | *Dominica* | Kew 25887 | ERS2203590 | 14.7 | 1.58 | 56 | 31.8 | 1.96 |
| *P. foliosa (2)* | *Venezuela* | HBV ORCH 07082 | ERS2203574 | 30.2 | 1.73 | 8 | 18.6 | 1.79 |
| *P. golungensis* | *Kenya* | Kew 17966 | ERS2203587 | 104.7 | 1.14 | 115 | 26.1 | 1.88 |
| *P. henrici | *Madagascar* | Kew 17856 | – | 22 | 1.52 | 56 | 7.7 | 2.04 |
| *P. humbertii (1)* | *Madagascar* | Fischer&Sieder FS2079 (WU) | ERS2203578 | 116.3 | 1.82 | 13 | 29.4 | 1.77 |
| *P. humbertii (2)* | *Madagascar* | Fischer&Sieder FS3017 (WU) | ERS2203580 | 35.1 | 1.93 | 11 | 34.2 | 1.89 |
| *P. melanantha* | *Kenya* | Kew 17954 | ERS2203584 | 207.2 | 0.94 | 50 | 12.2 | 1.8 |
| *P. modesta* | – | HBV ORCH 05165 | ERS2203573 | 56.8 | 1.35 | 13 | 31.7 | 1.93 |
| *P. odorata (1)* | *Nigeria* | Kew 17857 | ERS2203581 | 33.7 | 1.56 | 58 | 46.8 | 1.8 |
| *P. odorata (2)* | *Cameroon* | A. Russell 42 (YA) | ERS2203571 | 36.9 | 1.91 | 13 | 26.2 | 1.91 |
| *P. oreocharis (1)* | *Madagascar* | Fischer&Sieder FS2082 (WU) | ERS2203579 | 58 | 1.74 | 13 | 30.2 | 1.87 |
| *P. oreocharis (2) | *Madagascar* | Fischer&Sieder FS3152 (WU) | – | 20.2 | 1.82 | 13 | 10.5 | 1.8 |
| *P. paniculata (2) | *Cameroon* | L. Pearce 27 (YA) | – | 56.1 | 1.4 | 8 | 8.1 | 1.98 |
| *P. pinicola | *Barzil* | HBV ORCH 06606 | – | 31 | 1.11 | 7 | 9.2 | 1.98 |
| *P. steudneri* | *Kenya* | Kew 17956 | ERS2203585 | 10.6 | 1.45 | 58 | 33.5 | 1.87 |
| *P. tessellata (1) (=P.concreta)* | *Madagascar* | Kew 17859 | ERS2203582 | 176.2 | 1.1 | 58 | 32.7 | 1.89 |
| *P. tessellata (2) (=P.concreta)* | *Madagascar* | Kew 17860 | ERS2203583 | 216.1 | 1.1 | 49 | 33.3 | 1.93 |
| *P. tsinjoarivensis (1) | *Madagascar* | Fischer&Sieder FS3209 (WU) | – | 19.3 | 1.83 | 13 | 9.3 | 1.79 |
| *P. tsinjoarivensis (2)* | *Madagascar* | HBV FS4182 | ERS2203569 | 22.2 | 1.75 | 13 | 17.2 | 1.64 |

Notes:
Species analyzed; location of the collection and voucher; DNA concentration and purity before and after genomic library assembly.
* Species excluded due to low quality sequencing.

Additionally, fragments with a base repeated more than seven times in a row were avoided to reduce the capture of repetitive sequences present in many places in the genome. Finally, the reference sequences in blocks used for probe design totaled 63,720 bp and were brought together into a single FASTA file and sent to (MYcroarray, Ann Arbor, MI, USA) to produce the probes.

## Data generation

### Sonication and genomic library preparation

Extracted DNA was randomly fragmented by sonication using a Covaris S220 instrument (Covaris, Woburn, MA, USA), in order to evenly cover the full genome. Adapters were incorporated into the fragmented DNA using NEXTflex™ DNA Sequencing Kit and NEXflex™ Barcodes kit (BIOO Scientific, Austin, TX, USA). Uniquely indexed adapters were used for each sample. We selected fragments between 300 and 400 bp using Agencourt AMPure XP magnetic beads kit (Beckman Coulter, Brea, CA, USA). The genomic library was amplified following the program: 98 ºC for 2 min; 14 cycles (98 ºC for 30 s; 65 ºC for 30 s; 72 ºC for 60 s); 72 ºC for 4 min. The products were purified using a QIAquick PCR Purification Kit (Qiagen, Valencia, CA, USA). The genomic DNA concentrations before and after sonication and the amplification of the library were measured in a NanoDrop 2000c instrument (Thermo Fisher Scientific, Waltham, MA, USA) (Table 1) to ensure that the final concentration exceeded 400 ng/μL.

### Enrichment and sequencing

Before the enrichment, equimolar amounts (400 ng/μL) of each amplified library were pooled into six reactions, each one containing eight indexed samples. The enrichment method involves the selection of genomic regions and capture of DNA samples before sequencing (*Mamanova et al., 2010*). The enrichment was performed with MYBaits target enrichment system (MYcroarray, Ann Arbor, MI, USA), following the manufacturer's instructions. The probes were recovered using Dynabeads® MyOne™ Streptavidin C1 (Invitrogen Dynal AS, Oslo, Norway).

To increase DNA concentration, 14 cycles of PCR were performed for each hybridization reaction using Herculase II Fusion DNA Polymerase (Agilent, Waldbronn, Germany) and the following program: 98 ºC for 30 s; 14 cycles (98 ºC for 20 s; 60 ºC for 30 s; 72 ºC for 60 s), 72 ºC for 5 min. Sequencing was performed on the Illumina MiSeq platform (San Diego, CA, USA) by the Genomics Core Facility (University of Gothenburg, Sweden).

### Sequence editing

Illumina reads were processed using the program CLC assembly cell (CLC Bio, Aarhus, Denmark). Firstly, the Illumina adapter sequences were removed and low-quality reads were excluded. Reads were then mapped against the reference sequence used for probe design (*P. aphrodite*). Consensus sequences generated for each sample were converted into FASTA format using the SAMTools software (*Li et al., 2009*) using the mpileup tool with reference sequence option, allowing for the inclusion of indels in the consensus sequences. These sequences were used as a new individual reference sequences for each sample in a second round of mapping. Final consensus sequences were generated using mpileup, without the reference sequence option, to avoid erroneous base calling in low read-depth portions of the read alignment. Sequence alignment was performed using the auto strategy in MAFFT—Multiple Sequence Alignment Software Version 7 (*Katoh & Standley, 2013*) and later manually refined using Geneious Pro (Biomatters Ltd., Auckland,

New Zealand). In the last step we aligned the sequenced samples with the *Phalaenopsis aphrodite* subsp. *formosana* (NC_007499.1) chloroplast genome to obtain the sequenced region annotation.

## Alignment and phylogenetic analysis

### Hairpin inversions

Micro-structural features of chloroplast non-coding sequences can have a profound influence on the multiple sequence alignment, and hence also the phylogeny. Hairpins (short stem-loop structures in single stranded DNA or RNA), for example, can create sites that allow small inversions to occur at a high enough frequency that homoplasious inversions can be observed among sequences from closely related species (*Kelchner & Wendel, 1996*). Sometimes the inverted sequence is not so short and can disrupt phylogenetic analysis, leading to strongly supported but spurious groupings (*Joly et al., 2010*). Non-coding sequences, such as group II chloroplast introns, contain many such stem-loop structures (*Kelchner, 2002*).

We examined the non-coding sequences in our alignment for inverted (reverse complemented) sequences and tested for their effect on the phylogenetic inference. This was done by excluding all but one character of the inversion (to down-weight the inversion to a single event) and rerunning the analysis. The selected character to represent the inversion was arbitrarily chosen. This was done to avoid recoding the inversions as indel characters and creating a new, small partition (with only eight characters) that would have required many additional parameters, in comparison to our approach.

### Site exclusions

The alignment process can sometimes be confronted with small regions that are difficult to align, probably most often due to overlapping indel events. We identified several such regions and excluded them using the nexus block commands from the Bayesian analysis. Poor alignment excluded sites: 54459-54465, 55220-55224, 55484-55489, 67162-67167, 67426-67431, 68186-68192, 118838-118843, 119266-119364, 119493-119726, 120012-120019, 120867-120872, 121295-121393, 121522-121755, 122041-122048. Inverted loop-associated excluded sites: 74668-74669, 88120-88128, 90474-90481, 94193-94205, 94696-94699, 97241-97245, 104712-104716, 106631-106635.

The aligned sequences were partitioned based on the chloroplast annotation of 116 functional genes, seven pseudogenes and two partitions that concatenated untranslated regions—one partition containing introns with secondary structure and the other concatenating all intergenic sequences. The files that contained the introns were submitted to trimAl v1.2 (*Capella-Gutiérrez, Silla-Martínez & Gabaldón, 2009*) to test different gap deletion settings.

### Faster region assessment

We used the sequences from one sample on GenBank (FS1045) of *Polystachya cultriformis* and examined two published markers, *psbD-trnT* (870 length aligned to our samples) and *matK* (1,521 length aligned to our samples). We compared these sequences pairwise to one of our samples, *P. estrelensis*8, to check which was the faster-evolving

region. We then ran a MrBayes v.3.2.4 (*Ronquist et al., 2012*) analysis on the faster region, as a representative of a fast part of the chloroplast genome (fast cpDNA hereafter), with one copy of our dataset trimmed to this region alone plus the GenBank sequence.

We then ran successive analyses, using additional copies of the dataset interleaved in the same file, to explore the increase in support with an increase of characters evolving under the same model. This was to discover how much of the fast cpDNA data would be needed to achieve high support on most nodes in the phylogeny (i.e., among species but not necessarily within species). An important assumption we made at this stage was that the single fast region would contain mutations spread across most/all branches of the phylogeny. If this assumption was true, then a single region could carry changes representative of the entire phylogenetic history that we were exploring. The assumption is essentially one of i.i.d. (independently and identically distributed sites)—in that the sites would be representative of many unsampled sites and that double mutations would be rare—coupled with a sufficient dataset size to contain enough changes overall to reflect the history. Although the i.i.d. assumption is rarely true across sites, model-based analysis methods can cope with this, because the frequency of site patterns can be modelled by an i.i.d. process (*Steel & Penny, 2000*). So we are in effect mainly testing whether the original data set size was sufficient to carry changes reflective of the entire phylogeny under investigation.

### Random sample from all non-coding regions

A single region copied many times proved ineffective in recovering most nodes with support (see Results). We therefore explored using random samples of characters without replacement from among all of the non-coding regions in our dataset to test how much data from faster regions would yield supported trees across most nodes. We expected this approach to be less subject to the limitations caused by the stochastic nature of mutations coupled with the limited size of any one region. By sampling across many regions (over 58 kb in this case), even those few characters that have changed on short branches might be sampled occasionally. In contrast, a single region, by chance, may simply not contain any characters changing on a specific short branch.

We sampled without replacement 4%, 8% and 16% of the non-coding data using delete-fraction jackknifing in the seqboot program v3.69 (from http://evolution.genetics.washington.edu/phylip.html), excluding the poorly aligned parts and with down-weighting of the inverted loops (by excluding all but one character of each loop), in 20 replicates each. The approximate average (and range) of posterior probabilities (PP) per clade was taken across the 20 replicates to get an indication of the likely support for selected clades that a non-coding dataset of these sizes would generate. These values were plotted on the whole alignment analysis to compare to the support received when using the whole dataset. Given that the largest dataset we used here (hereafter the "16% dataset," or ~9.2 kb) failed to recover support for all nodes found in the whole genome analysis (see Results), we did not end up analyzing the smaller replicates.

### Phylogenetic analysis

MrBayes v.3.2.4 analyses were used for phylogenetic inference. These analyses were run for five million generations (two million for the random sample replicates), using a mixed substitution model (plus gamma and invariant sites) to account for among-site rate variation. Priors on branch lengths were set to unconstrained: exponential (100) to minimize the chance of inferring incorrectly long branches (*Marshall, 2010*), otherwise with default settings. The paired runs were checked for convergence and high effective samples sizes in the MrBayes output and Tracer v.1.6 (*Rambaut et al., 2014*), respectively. Burn-in generations were removed by discarding 10% of the samples of parameters and trees, while summarizing in TreeAnnotator v.1.8 (*Rambaut & Drummond, 2010*) to ascertain clade PP. Trees were rooted using the *Phalaenopsis* sequence. Analyses using the character partitions were also done, returning nearly identical results to the analysis described above, so they are not reported further.

## RESULTS

Our NGS approach allowed the capture of coding and non-coding regions throughout the chloroplast genome. We recovered approximately 132 kb, after the exclusion of gaps, representing 116 genes, seven pseudogenes, as well as regions with intergenic sequences and introns with secondary structure. Compared to the reference annotation, seven genes contained frameshifts that are usually associated with pseudogenization and corresponded to previously reported pseudogenized genes in orchids (*Luo et al., 2014*).

We excluded eight of the 48 samples due to the low quality of the sequencing results (Table 1). These eight samples showed lower DNA concentrations after the genomic library construction assembly, which may be the cause of low quality sequencing. The remaining 40 samples were submitted to the EMBL/ENA database under accession numbers ERS2203551–ERS2203590. The coding regions have 48,308 polymorphic sites (38.4%). Introns with secondary structure and regions with intergenic sequences have 21,264 (16.9%) and 56,226 (44.7%) polymorphic sites, respectively. The alignment of the concatenated data showed an unbalanced (but fairly typical) mean nucleotide composition of $A = 29.9\%$, $C = 19.9\%$, $G = 19.4\%$ e $T = 30.8\%$.

### Analysis results

#### Hairpin inversions

In the non-coding part of alignment, we found evidence for eight putative small inversions (Table 2), based on the presence of inverted repeated motifs that could form stems at least 4 bp long. Stems of this length or longer are part of models of group II structures (*Michel, Kazuhiko & Haruo, 1989*; *Toor, Hausner & Zimmerly, 2001*; *Kelchner, 2002*) and are consistent with sequence patterns observed by one of us in the *rpL16* intron sequences in other taxa (*Pfeil et al., 2002*).

#### Faster region assessment

The pairwise identity between *Polystachya concreta*5 and *P. concreta*8 (whose common ancestor is relatively old and near the crown of *Polystachya*) for *psbD-trnT* was 98.7%.

**Table 2  Alignment co-ordinates and sequences of putative stem-loop structures that appear to have undergone inversion in *Polystachya*.**

| Loop | Alignment co-ordinates | Sequence (majority) | Sequence (minority) | Substitutions before/after inversion reverted | Sequences (minority) |
|---|---|---|---|---|---|
| A | 74,660–74,678 | TCTATCTA-GAA-TAGATAGA | TCTATCTA-**TTC**-TAGATAGA | 3/0 | bicolor, concreta7, concreta9, tesselata1, melanantha, Phalaenopsis |
| B | 88,106–88,142 | GGCCCAATCTTT$_C$-TTTTTTTGAGGA-AAAGATTGGGCC | GGCCCAATCTTT$_C$-**TCCTCAAAAAAA**-AAAGATTGGGCC | 8/0 | humbertii2, oreocharis1, tsinjoarivensis2, melanantha |
| C | 90,456–90,496 | AGTAAGAACTCAGCG-GGGTAAGGCCT-CGCTGAGTTCTTACT | AGTAAGAACTCAGCG-AGG**CCTTACCC**-CGCTGAGTTCTTACT* | 7/0 | eurychila |
| D | 94,175–94,222 | ATTGAAGTAATGAGCCC-CAAGATGAATATGA-GGGCTCATTACTTCAAT | ATTGAAGTAATGAGCCC-**TCATATTCATCTTG**-GGGCTCATTACTTCAAT | 8/0 | humbertii1, steudneri |
| E | 94,679–94,717 | GTATCTAAGGAAGATCC-AAAGG-GGATCTTCCTTAGATAC | GTATCTAAGGAAGATCC-**CTTCT**-GGATCTTCCTTAGATAC | 5/2 | estrelensis12, estrelensis13, estrelensis14, eurychila, concreta7, concreta8, golungensis, oreocharis1, melanantha, humbertii1, Phalaenopsis |
| F | 97,230–97,258 | AACGTCCAGTG-CCAAAGT-CACTGAATGGG | ^CCCATTCAGTG-**ACTTTGG**-CACTGAATGGG | 6/0 | All 13 sequences of the estrelensis clade (cartooned in Figs. 2 and 4) |
| G | 104,698–104,730 | ATTC$_A$ATTCTTC-ATTCTTTCAA-GAAGAATGAAT | ATTC$_A$ATTCTTC-AT**GAAAGAAA**-GAAGAATGAAT | 6/0 | foliosa1, foliosa2, odorata1, odorata2, Phalaenopsis |
| H | 106,618–106,650 | ATTCATTCTTCAT-GAAAGA-ATGAAGAAT$_T$GAAT | ATTCATTCTTCAT-**TCTTTC**-ATGAAGAAT$_T$GAAT | 6/0 | foliosa1, foliosa2, odorata1, odorata2, Phalaenopsis |

**Notes:**
The loops that appear to have undergone inversions are underlined. Bolded bases would be treated as phylogenetically informative if the alignment does not take into account the inversion. Bases not involved in stem formation are in subscript.
* Occurs in only one sequence.
^ Only the sequences with the inversion appear to form the stem in this case.

The pairwise identity for these samples for *matK* was 99.2%. *PsbD-trnT* was therefore used as the representative fast cpDNA region. The analysis with a single copy of this dataset (870 bp) yielded a MCC tree (Fig. 2A) with only five nodes with high support (>0.95 PP). Increasing the number of copies did not result in much improvement. The analysis using 16 copies of *psbD-trnT* (~14 kb) produced a MCC tree (Fig. 2B) containing only eight highly supported nodes.

### Random sample from all non-coding regions

The pairwise identity between *Polystachya concreta*5 and *P. concreta*8 was 98.7% across 58 236 bp of non-coding region contained in our alignment. This compares to 98.8% identity between the same samples across the coding regions in our alignment.

The replicate datasets that sampled 16% of the original non-coding alignment (excluding poorly aligned parts and down-weighting the inverted loops) failed to return all nodes/clades found in the whole genome analysis. Of 13 selected nodes found in the tree from the whole genome (four subtended by relatively long branches, four by medium length branches, and five by short branches), only five were found with high support across most or all replicates (i.e., at least 16 of 20 replicates had ≥0.95 PP). Three of the selected nodes instead had five or fewer replicates with high support (≥0.95 PP), but only one or no replicates that contained highly supported contradictory nodes (thus the support for the expected node was ≤0.05). Finally, five of the nodes had generally poor support among replicates (i.e., five or fewer replicates had ≥0.95 PP along with six or more replicates with ≤0.05 PP for these nodes). Sixteen percent of datasets (~9.2 kb) were recovered from seven to 21 highly supported nodes among replicates (mean = 14.8), with more nodes recovered in 19 of 20 replicates than was the case with the larger repeated *psbD-trnT* dataset (~14 kb and eight supported nodes). This character sampling strategy was probably more reflective of the underlying support for various nodes than using repeated copies of a single small dataset.

The mutually exclusive foliosa1/concreta2 versus foliosa1/foliosa2 clades (see below) were also examined in the 16% datasets. In the first case (foliosa1/concreta2), just four replicates contained this clade with high or moderate support (≥0.90 PP). The contradictory second grouping (foliosa1/foliosa2) was found with a similar level of support (≥0.90 PP) in only two replicates. The fact that both groupings could be recovered, with high support, in at least some replicates suggests that the original dataset contains the signal of both clades. A NeighborNet analysis (Fig. 3B) confirmed that a mixture of patterns exists in the original dataset involving foliosa1, foliosa2, concreta1 and concreta2.

## Phylogenetic analyses

Analyses with and without the inverted loops (the latter by down-weighting to a single character) returned almost identical trees. The results of only the latter analysis is presented in this section. The tree we recovered was able to resolve the phylogenetic relationships among the groups of the large clade selected for this study, with high support values on almost every node (Fig. 3). The tree was characterized by a large clade with

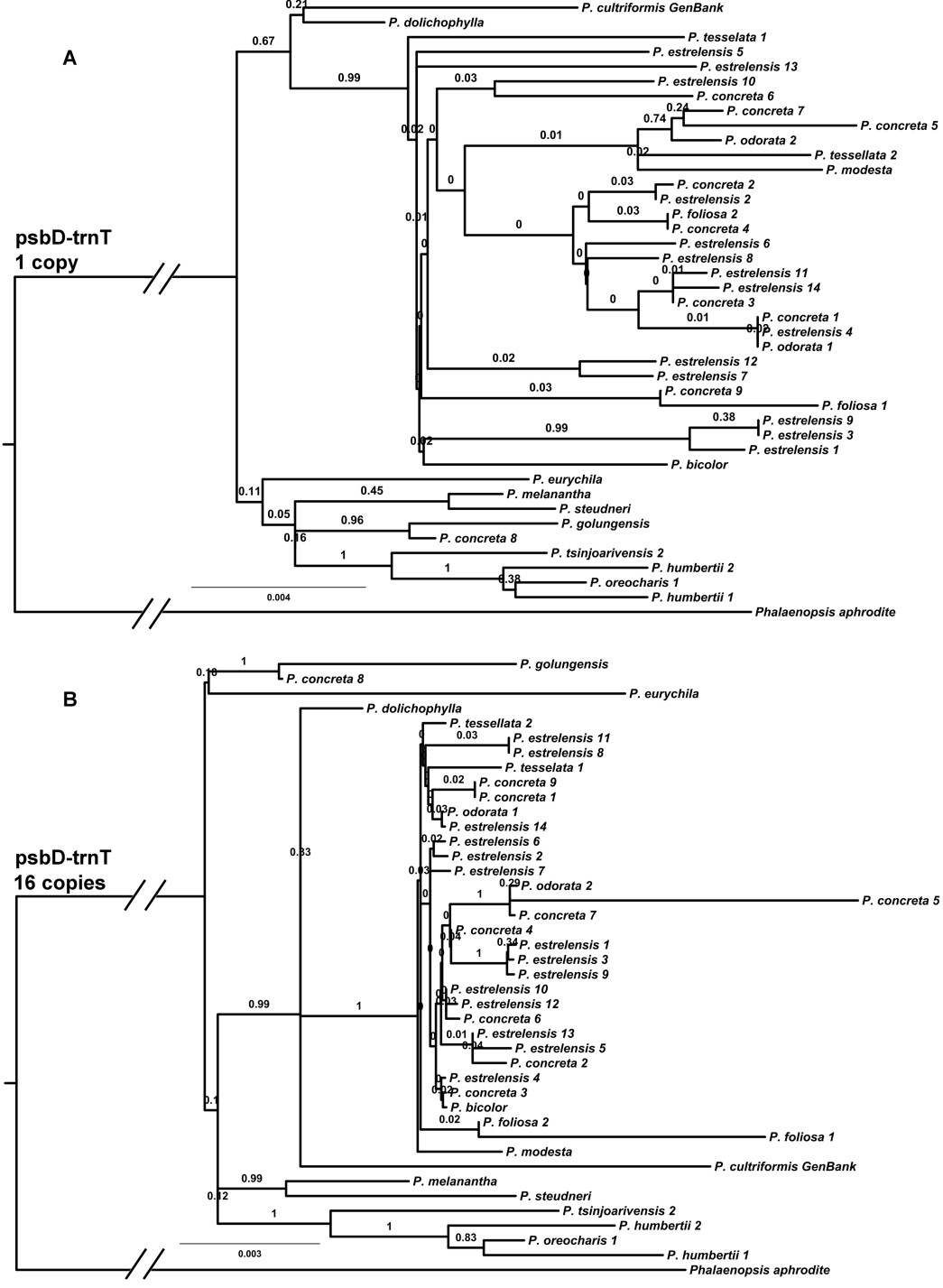

**Figure 2 Phylogeny of the psbD-trnT region estimated using Bayesian analysis and rooted using *P. aphrodite*.** Posterior probabilities are show above branches. Scale bar is in substitutions per site. The two branches leading to the root have been foreshortened to reduce space and are thus not to scale. (A) Phylogeny based on a single copy of *psbD-trnT*. (B) Phylogeny based on 16 identical copies of the *psbD-trnT* data set.

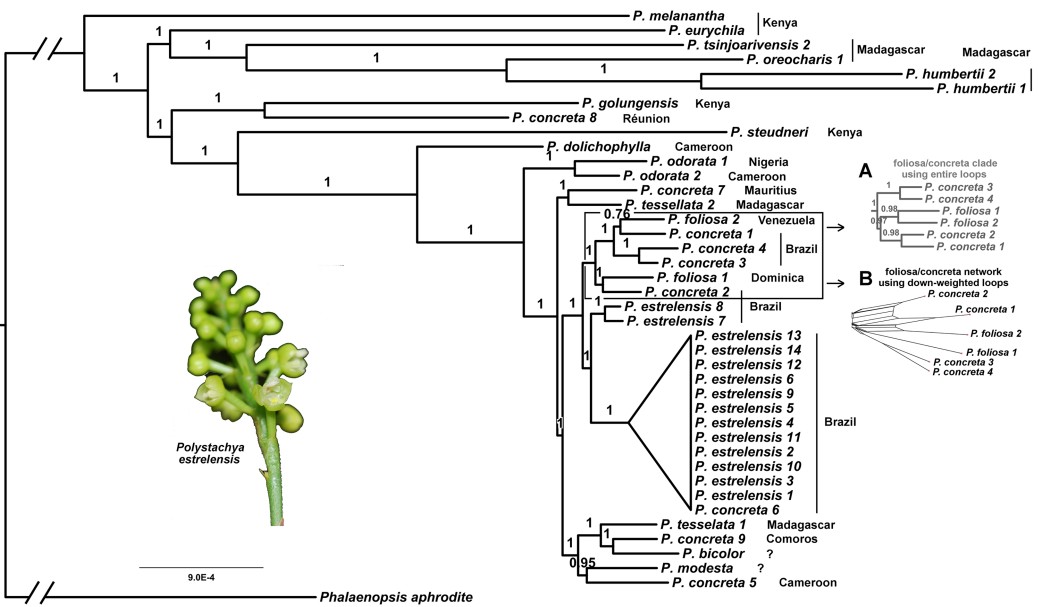

**Figure 3 Plastid phylogeny from Bayesian analysis rooted using *P. aphrodite*.** Posterior probabilities are show above branches. The *Polystachya estrelensis* group has been collapsed to reduce detail. Scale bar is in substitutions per site. The two branches leading to the root have been foreshortened to reduce space and are thus not to scale. The two insets are not at the same scale as the main figure. *Main Figure*: phylogeny based on the data set with poorly aligned regions excluded and loops down-weighted. (A) (in gray) Phylogeny of the foliosa/concreta group based on the full inclusion of loops. (B) (in black) NeighborNet network of the foliosa/concreta group based on the down-weighted loops. *P. estrelensis* photo credit: N. Lopes de Abreu.

relatively short branches containing only sequences from the Neotropics, with a grade of a few small clades and single sequences containing the remaining sequences (Fig. 3). The large clade contained 21 sequences from Brazil, Dominica and Venezuela, whereas the grade included 19 sequences from tropical central and eastern Africa, as well as Madagascar and the nearby islands (Fig. 3).

The grade recovered include a few geographically identifiable clades (Fig. 3). One of these, attaching fairly deeply within the crown, consists of four Malagasy sequences (*Polystachya humbertii*1, *P. humbertii*2, *P. oreocharis* and *P. tsinjoarivensis*2) that are sister to a Kenyan sequence (*P. eurychila*). Another clade comprises a Kenyan sequence (*P. golungensis*) and one from Reunion (*P. concreta*8). A third clade contains a pair of central African sequences, one from Cameroon (*P. odorata*2) and one from Nigeria (*P. odorata*1). A fourth clade contains sequences from central Africa (*P. concreta*5 from Cameroon), Madagascar (*P. tesselata*1), the Comoros (*P. concreta*9) and two sequences without certain provenance. Finally, a fifth pair of sequences were from samples collected from Mauritius (*P. concreta*7) and Madagascar (*P. tesselata*2). Lineages containing only a single sequence in this grade included samples from Kenya (*P. melanantha* and *P. steudneri*) and Cameroon (*P. dolichophylla*).

Sequences from the widely sampled and widely distributed *P. concreta* did not form a monophyletic group and occurred on different branches of the tree, separated by

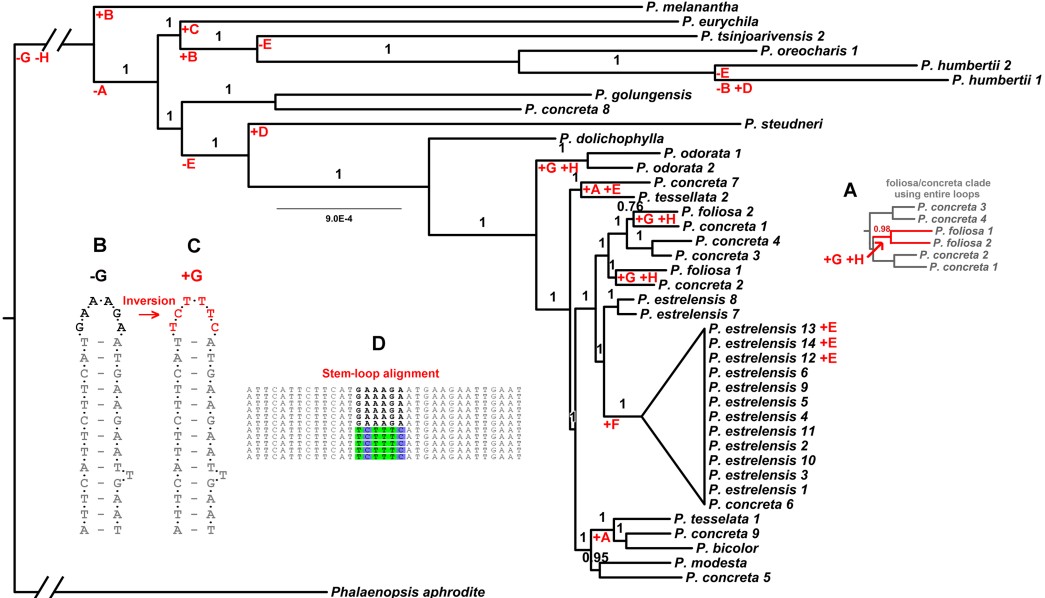

**Figure 4 Parsimonious gains and losses of non-coding loop inversions in *Polystachya* relative to the outgroup sequence, *P. aphrodite*, mapped on to the plastid phylogeny.** The letter codes designate loops as per Table 2. *Main Figure*: phylogeny estimated using down-weighted loops (from the main panel, Fig. 1). Where equally parsimonious interpretations were possible, accelerated transformation has been used. (A) Part of the phylogeny estimated using entire loops for the foliosa/concreta group. The mapping of gains of two loop inversions shared by foliosa1 and foliosa2 on this topology is in contrast to the mapping on the topology using down-weighted loops (main figure). (B) A diagrammatic representation of the stem-loop structure with the majority form of the loop sequence (in black). (C) The stem-loop structure with the proposed inversion of the loop sequence (in red). (D) The consequence on the alignment before down-weighting of the loop sequence (loop sequence in bold— majority form; loop sequence with back colors—inverted form).

several well supported nodes (Fig. 3). Similarly, the two *P. tessellata* sequences from Madagascar did not form a clade. *Polystachya estrellensis* sequences form a clade with *P. concreta* sequences collected in Brazil. Although the sequences of *P. estrellensis* are thus paraphyletic, whether the taxon itself is paraphyletic cannot be established for certain here. The identification of this *P. concreta* sample could be wrong, given that the identification of these species is confused in Brazil and sometimes they are considered synonymous (see also Discussion).

### *With versus without loops*

The down weighting of the inversions we identified (by excluding all but one character per inversion) resulted in a similar, but not identical, phylogenetic inference. The differences among the maximum clade credibility (MCC) trees involved *P. foliosa1*, *P. foliosa2*, *P. concreta1*, *P. concreta2*, *P. concreta3* and *P. concreta4*. The analysis with the inversions included returned this tree (Figs. 3A and 4A):

((concreta3,concreta4):1,((foliosa1,foliosa2):0.98,(concreta1,concreta2):0.98):0.97)

with clade PP listed after each node. In contrast, the inference resulting from down-weighted inversions returned this tree (Fig. 3 main panel and Fig. 4 main panel):

((foliosa1,concreta2):1,((concreta3,concreta4):1,(foliosa2,concreta1):0.76):1)

There are several supported differences between these trees, with at least one corresponding to the way the inverted loops are weighted. When the entire loops are analyzed, *P. foliosa1* and *P. foliosa2* are supported as sisters, with these two sequences appearing to share two loop inversions (if this topology is correct; Fig. 4A). However, down-weighting the inversions produces a tree consistent instead with two independent inversions (Fig. 4 main panel).

## DISCUSSION

### Chloroplast genome sequence provides a robust phylogeny

In this work we used the nearly complete chloroplast sequences of 40 *Polystachya* samples to infer a robust plastid phylogeny. The dataset significantly increased the phylogenetic resolution within the genus. Thus, our results suggest that increasing the number of molecular markers has the potential to solve not only the relationships among species, but also to identify new *Polystachya* clades and define new sections. The delimitation of new sections will, however, depend upon the inclusion of more taxa than was done by this study—in other words a higher coverage of the genus. Below we highlight some of the clades recovered, their morphological and/or geographical characterization, and a comparison with previous studies.

### *Polystachya bicolor/rosea position contradicts Russell's Clade III*

Of the two species selected as possible distant relatives to provide more context in the phylogenetic inference, one of them, *P. bicolor* (=*P. rosea*), appears in a clade together with samples of *P. concreta* (from Cameroon and from the Comoros), *P. tesselata* (= *P. concreta*) and *P. modesta*. Not surprisingly, the clade that includes *P. bicolor/rosea* is deeply nested within the ingroup, thus contradicting the monophyletic Clade III presented by *Russell et al. (2010b)*.

In prior studies, *Polystachya bicolor/rosea* has an uncertain position in the phylogenetic trees. In an analysis using plastid markers and Bayesian inference, this species appears in a large polytomy with *P. concreta* and other related species (*Mytnik-Ejsmont, 2011*), or related to species of other clades (*Russell et al., 2010b*) depending on the marker used. A phylogeny using nuclear data (ITS sequences) highlighted the lack of monophyly of this species (*Mytnik-Ejsmont, 2011*), which may be connected to the difficulty in identifying it. *Polystachya bicolor/rosea* is often mistaken for *P. concreta*, since differentiation between these is made by subtle differences in the shapes of leaves, and the size and color of the flowers. Unlike *P. concreta*, which has a pantropical distribution, *P. bicolor/rosea* is restricted to Madagascar, Comoros and the Seychelles (*Mytnik-Ejsmont, 2011*).

### Brazilian sequences form a clade

The monophyletic nature of the group formed by the Brazilian accessions, contrasting with the paraphyletic group made up of African accessions, is consistent with the hypothesis that Africa is the center of origin with a subsequent (i.e., more recent) dispersal into the Neotropics (*Russell et al., 2010a*, *2010b*).

### Hybrid origins of some taxa suggested

The hybrid origin of *P. concreta* is a possible explanation for this species being found in different positions in the tree (*Russell et al., 2010b*). *P. concreta* individuals that have dispersed out of Africa are tetraploid, whereas plants found in continental Africa can be diploid or tetraploid. The sister taxa of African *P. concreta* are diploid (*Russell et al., 2010b*), indicating that tetraploidy is a derived state in *P. concreta*. Allotetraploidy in *P. concreta* has been confirmed by analysis of low copy nuclear genes (*Russell et al., 2010a*).

Interspecific hybridization events, as in *P. concreta*, are considered a source of chloroplast genome exchange via introgression. Chloroplast genome exchange among species is sometimes suggested as an explanation for the inconsistencies between phylogenetic trees based on nuclear and plastid markers in, e.g., *Populus* (Salicaceae) (*Smith & Sytsma, 1990*; *Tsitrone, Kirkpatrick & Levin, 2003*), *Nothofagus* (Nothofagaceae) and Crassulaceae (*Mort et al., 2002*; *Acosta & Premoli, 2010*). In *Nothofagus*, chloroplast capture results in the association of chloroplast genomes with geographic locations, rather than taxonomic relationships (*Acosta & Premoli, 2010*). Relationships based on geographic location could be explored as a possible explanation for the proximity of *P. concreta* (accesses from Brazil) with *P. estrellensis* (also from Brazil) and not with non-Brazilian accessions of *P. concreta*. In this case a study of nuclear markers of these taxa would be needed.

### Neotropical species

Relationships in the group that includes *P. concreta*, *P. foliosa*, *P. estrellensis* and other species are not well resolved due to the low sequence divergence levels between species found in both plastid and nuclear genes (*Russell et al., 2010a*, *2010b*, *2011*; *Mytnik-Ejsmont, 2011*). Generally, the morphological variation observed in this group is identified as *P. concreta*. Although *P. estrellensis* is considered a valid species on the official plant list of Brazil (*Barros et al., 2010*), there is no consensus on synonymy with *P. concreta*. This can be seen in the herbarium identifications that sometimes consider them as two distinct species, but sometimes as the same species. The same occurs with *P. foliosa*, a name which would only be correctly applied to plants from the Amazon basin, the Guyana Shield and the West Indies (*Peraza-Flores, Fernández-Concha & Romero-González, 2011*). This circumscription is not accepted by *Mytnik-Ejsmont (2011)*, who considers *P. estrellensis* and *P. foliosa* to be synonymous.

Genetic dissimilarity between African and Neotropical tetraploids was reported by *Russell et al. (2010a)* and *Russell et al. (2011)*, but the delimitation *P. estrellensis*, *P. concreta* and *P. foliosa* remained uncertain. According to our results, under a molecular perspective, *P. estrellensis* should be considered distinct from *P. concreta*. Moreover, our results do not

corroborate the placement in synonymy of *P. estrellensis* and *P. foliosa* as proposed by *Mytnik-Ejsmont (2011)*. In our tree *P. foliosa* forms a highly supported group with some *P. concreta* sequences (from samples collected in Brazil). Finally, although our results indicate a possible separation of Brazilian and African *P. concreta*, the delimitation of this species remains uncertain, considering that there is no generic taxonomic revision that has rigorously analyzed the morphological variation in this species. Moreover, considering the reticulated evolution by *Russell et al. (2010a)*, further investigation with nuclear markers would be necessary.

Taken together, our analysis suggests that *P. estrellensis* can be considered a distinct species from *P. concreta* and *P. foliosa*, and that Brazilian and African *P. concreta* should probably be treated as different species. Evidence of hybridization influencing the evolution of *P. concreta* (*Russell et al., 2010a*, *2010b*) highlights how importance it will be to also consider bi-parentally inherited nuclear DNA when inferring of phylogenetic relationships between this species and other species of the genus. The placement in synonymy of *P. estrellensis* and *P. foliosa* proposed by *Mytnik-Ejsmont (2011)* was not confirmed by this study. In our results, *P. foliosa* forms a highly supported clade including Brazilian samples of *P. concreta*.

## Implications for data requirements
### The entire chloroplast is more useful than a fast subset
By using a relatively large number of chloroplast sequences we were able to resolve the polytomy involving the Neotropical species. But, if on one hand this dataset is promising in the formulation of more robust phylogenetic hypotheses, on the other hand, the complete chloroplast genome sequencing may be costly for systematic projects that consider genera with many species (*Särkinen & George, 2013*), such as *Polystachya*, which has about 250 species. This was the main motivation for testing how well a smaller sample of the chloroplast genome would perform in phylogenetic inference. This was done in two ways: by duplicating a fast region and analyzing multiple copies of this dataset, and by sampling without replacement from all non-coding regions in our alignment.

We found that sampling without replacement up to ~9 kb of non-coding sequence (16% of our alignment) was not sufficient to return a robust inference across all nodes. This was in contrast to the analysis of the entire chloroplast and showed that in the case of these samples of *Polystachya*, more data were needed to resolve their relationships. The cost of primers, amplification and Sanger sequencing of only three or four regions begins to exceed that of gene capture of the entire chloroplast. It is therefore more cost effective and produces a more robust result to undertake the collection of the entire chloroplast genome. That said, our 16% sample did resolve some nodes with high support, and other nodes obtained moderate to high support from a few of the replicates. This suggests that these data are on the way to resolving most nodes, but a gradual increase in resolving power occurs as characters are added.

Duplicating a single fast region even 16 times, in this case *psbD-trnT* copies totaling ~14 kb, failed to achieve a robustly resolved phylogeny. The results for the *psbD-trnT*

duplicated analysis was poorer even than that of sampling fewer but more representative characters across the non-coding region (above). It appears that a small sample size (only 870 bp of independent sequence sites) is a serious source of stochastic error in this case. Sampling one versus 16 copies of the same dataset only slightly increased the number of resolved nodes (but still falling short of the number of nodes usually resolved with support by the smaller 16% sample), confirming the limitations of the original dataset. It is likely that the original dataset simply did not contain sites that changed on most branches of the phylogeny during the span of history that we investigated.

Numerous previous studies have also examined which regions of the plastid genome are typically evolving faster than others (*Small et al., 1998*; *Shaw et al., 2005*, *2007*, *2014*). Prior to NGS methods, the aim was to identify the "best" regions, when sequencing only a limited number could be afforded in most projects. However, given current technology, we should shift our focus to whether a few of the "best" regions are cost effective compared to using the entire genome, as the latter has become affordable for even small phylogenetic projects.

### Homoplastic hairpin inversions affect phylogenetic analysis

One issue raised here that is rarely taken into account in analyses of whole chloroplasts is that sequence patterns at the small scale, namely hairpin inversions of loops, can still have an effect on phylogenetic inference, despite using very large data sets. Our results indicate that at least some of the differences between the trees inferred using entire loops versus down-weighted loops were driven by these hairpin loop inversions. This kind of phylogenetic effect has been observed in other cases, although with smaller data sets (*Kim & Lee, 2005*; *Joly et al., 2010*). If loops invert in a single molecular event (as is currently believed: *Kelchner & Wendel, 1996*; *Kim & Lee, 2005*), such as an intra-molecular recombination, then there is no good reason to use each character state difference found between sequences in the entire loop in an analysis. This simply inflates the phylogenetic impact of a single event, treating it instead as many independent events (corresponding to the number of character state differences in the inversion), as also noted by *Kim & Lee (2005)*. As shown here, a larger data set simply does not give license to ignore known analytical pitfalls.

Together, these findings show that sampling the entire chloroplast, analyzed carefully, is a better option than sampling a few (even a dozen or more) fast regions. This is true, at least in *Polystachya*, but a similar result has also found by other studies, such as *Parks, Cronn & Liston (2009)* for *Pinus*. Based on cost alone, it seems there is no benefit to be gained by screening the chloroplast for faster markers when there are many short branches in the particular tree, as there are here. Whole chloroplast analyses are likely to be a better way forward than sampling individual chloroplast markers in addressing many phylogenetic questions. If gene capture is used, as it was here, it is also very easy to add probes to unlinked nuclear regions, further increasing the power of this approach as a general solution to the issue of data sampling.

## CONCLUSION

Our results show that significantly increasing the number of nucleotides can be an effective option in the phylogenetic inference of taxonomic challenging taxa, such as the orchid genus *Polystachya*. We generated complete chloroplast sequences of 40 *Polystachya* specimens using a combination of Illumina NGS sequencing and a sequence capture, which solved a notorious polytomy for Neotropical species. Our tests on how well a smaller sample of the chloroplast genome would perform in phylogenetic inference shows that the whole chloroplast is a better option than selecting just a few highly variable markers. Full plastid genomes appear particularly powerful when the expected tree is likely to contain many short branches, but nonetheless need to be analyzed with care.

## ACKNOWLEDGEMENTS

We thank our colleagues in our research groups for discussions, advice and support throughout this project. We also thank the anonymous reviewers whose comments helped improve the paper.

### Funding

Alexandre Antonelli is supported by grants from the Swedish Research Council (B0569601), the European Research Council under the European Union's Seventh Framework Programme (FP/2007-2013, ERC Grant Agreement n. 331024), the Swedish Foundation for Strategic Research, the Knut and Alice Wallenberg Foundation, the Biodiversity and Ecosystems in a Changing Climate programme, the Wenner-Gren Foundations and the David Rockefeller Center for Latin American Studies at Harvard University, and the Faculty of Science at the University of Gothenburg. There was no additional external funding received for this study. The funders had no role in study design, data collection and analysis, decision to publish, or preparation of the manuscript.

### Grant Disclosures

The following grant information was disclosed by the authors:
Swedish Research Council: B0569601.
European Union's Seventh Framework Programme: FP/2007-2013, ERC Grant Agreement n. 331024.
Swedish Foundation for Strategic Research.
Knut and Alice Wallenberg Foundation.
Biodiversity and Ecosystems in a Changing Climate programme.
Wenner-Gren Foundations.
David Rockefeller Center for Latin American Studies at Harvard University.
Faculty of Science at the University of Gothenburg.

## Competing Interests

The authors declare that they have no competing interests.

## Author Contributions

- Narjara Lopes de Abreu conceived and designed the experiments, performed the experiments, analyzed the data, prepared figures and/or tables, authored or reviewed drafts of the paper, approved the final draft.
- Ruy José Válka Alves conceived and designed the experiments, contributed reagents/materials/analysis tools, authored or reviewed drafts of the paper, approved the final draft.
- Sérgio Ricardo Sodré Cardoso conceived and designed the experiments, contributed reagents/materials/analysis tools, authored or reviewed drafts of the paper, approved the final draft.
- Yann J.K. Bertrand analyzed the data, authored or reviewed drafts of the paper, approved the final draft.
- Filipe Sousa performed the experiments, analyzed the data, authored or reviewed drafts of the paper, approved the final draft.
- Climbiê Ferreira Hall analyzed the data, authored or reviewed drafts of the paper, approved the final draft.
- Bernard E. Pfeil analyzed the data, prepared figures and/or tables, authored or reviewed drafts of the paper, approved the final draft.
- Alexandre Antonelli conceived and designed the experiments, contributed reagents/materials/analysis tools, authored or reviewed drafts of the paper, approved the final draft.

## Field Study Permissions

The following information was supplied relating to field study approvals (i.e., approving body and any reference numbers):

Permits to collect were provided by the Ministério do Meio Ambiente (MMA), Instituto Chico Mendes de Conservação da Biodiversidade (ICMBio) and Sistema de Autorização e Informação em Biodiversidade (SISBIO) (29478-1).

## DNA Deposition

The following information was supplied regarding the deposition of DNA sequences:

DNA Sequences have been deposited at EMBL/ENA under accession numbers ERS2203551–ERS2203590.

## Data Availability

The sequence alignment files are provided as Supplemental Files.

## Supplemental Information

Supplemental information for this article can be found online at http://dx.doi.org/10.7717/peerj.4916#supplemental-information.

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
