# Peer review of "The use of chloroplast genome sequences to solve phylogenetic incongruences in Polystachya Hook (Orchidaceae Juss)"

_PeerJ, doi:10.7717/peerj.4916_

## Round 0.1 · original submission · Major Revisions

The three reviewers performed a very thorough review of the manuscript and I agree with all of their suggestions, by doing so, the manuscript can be greatly improved. I strongly suggest authors to analyze all advice given by the reviewers and in case of refusing to perform any of the suggested changes, it needs to be clearly explained.

Reviewer 1 ·

Basic reporting

The authors cite one enrichment method (i.e. Anchored Hybrid Enrichment), their citations could be enriched by including other published studies applying this method to other plant genera. Also, authors should notice that the Anchored Hybrid Enrichment method targets nuclear loci and that a revision of equivalent techniques focused on chloroplast markers is missing in their introduction.

Experimental design

1. I have some concerns about the logic behind the subsampling schemes tested. First, it is not clear to me why the authors having sequenced and annotated 40 full chloroplast genomes they limited their search of potentially fast evolving regions to two previously published conventional markers. There is no explanation about why the authors expect that analyzing multiple copies of the same region will have a positive impact on phylogenetic reconstruction. Even if economic resources are limited, researches make an effort to include as many independent loci as possible. With the data that the authors have on hand they could being offering a more meaningful subsampling scheme where different regions (e.g. genes, introns and intergenic spacers) are ranked according to their phylogenetic informativeness (see Townsend 2007 method) before designating subsets of “fast evolving” regions. This procedure would allow the comparison of several independent fast evolving regions for a better evaluation of how many of these data is necessary to obtain a phylogenetic tree with comparable resolution and support to that obtained from the analysis of the full data set. Second, fast evolving sites or regions tent to provide support for shallow nodes, whereas slower evolving regions can provide support for deeper nodes. Therefore, excluding slower evolving regions can result on low resolution and/or support for deep nodes. Third, unusually fast evolving regions or sites can introduce phylogenetic noise which can decrease the resolution and/or support obtained. This latter possibility is not considered or discussed in the manuscript.

2. Inversions were reduced to a single randomly selected site within them to avoid overweighting this evolutionary event during the phylogenetic analyses. The authors should consider that the primary homology assumption for that selected site is still incorrect (i.e. they shouldn´t be aligned at the same position) and that in the case that a substitution event followed the inversion, the potential groupings resulting from their analysis can be confused.

3. Taking into account the approximate average and range of PP for comparing the randomly sampled matrices is a way of comparing them, but it does not consider the topology.

4. For probe design the plastome sequence of Phalaenopsis aphrodite subsp. formosana was divided into 360 bp long blocks, please specify if they were or not overlapping blocks.

5. Two species are cited as outgroup, however Phalaenopsis aphrodite was used for rooting the trees. Why P. aphrodite is not cited as outgroup?

Validity of the findings

1. In the conclusion section: you only generated 40 Polystachya specimens full plastome sequences, not 48.

Additional comments

The authors generated and analyzed a comprehensive amount of plastid DNA data (fully sequenced genomes) for improving our knowledge on the phylogenetic relationships of a lineage within the orchid genus Polystachya. They tested different subsampling schemes of presumably fast evolving regions in order to estimate how much of these data is needed to obtain a well resolved and supported phylogenetic hypothesis and compared this against their tree resulting from the analysis of the full data set. The probe capture strategy applied is valuable as they demonstrated that a full chloroplast genome could be assembled from the retrieved regions.

In addition to these general comments I have some particular comments that can help the authors improving their manuscript.

1. I suggest to narrow the title down to the genus Polystachya since even within it they only targeted a single lineage.
2. Otherwise indicated by the journal rules, all names under the genus level must be written in italics, this includes names of genus sections.
3. Generally, gene names are written in italics.

Reviewer 2 ·

Basic reporting

Although no easy to read, the author used the correct English expression in manuscript. And I can't found mistakes in other parts including reference, figs and tables. The result is self-contained.

Experimental design

Some doubts should be resolved, but no so fatal. The details listed in General comments for the author.

Validity of the findings

no comment.

Additional comments

L73-74: more reference should be add here since this number isn't the latest one.
L135: I don't think that is complete one, "nearly complete" would be better.
L161: Please explain here why use that as outgroup(ex. "a section far away based on XX's taxonomic treatment").
L162-165: Actually I found your outgroup should be Phalaenopsis, isn't it? You use the sequence from that to root the tree so phalaenopsis was the outgroup indeed.
L179-180: Why seven?
L255-265: I am confused why you used these two regions and the P. estrelensis8 as representative? And only psbD-trnT and matK have been estimated here, I don't think it can be the representative of chloroplast genome.
L266: As the reference is Phalaenopsis, the non-coding regions are no credible in Polystachya based on your method since these regions are highly influenced by leading reference. Moreover, these regions are various among species especially in different taxonomic tribe. Any confuse non-coding regions without steable estimation should be removed from the downstream phylogenetic reconstruction.
L415-418: The P. bicolor was the outgroup species, it was inpossible to nest within ingroup as soon as you root. Also it is not suitable to use the species in confusion as outgroup.
L432-433: Paraphyletic group is not a evidence for high species diversity.
L436: Hybrid origin is not a correct word since no evidence to distingush that from incomplete lineage sorting, alternatively the reticular evolution could be used.

Reviewer 3 ·

Basic reporting

.

Experimental design

.

Validity of the findings

.

Additional comments

1. The name of genus Polystachya should better be added in the title of the article for a clear expression.

Next generation sequencing commonly indicates more than one technique, the Next generation sequencing technique is just a tool, that enabled faster collection of sequence data, and did not contribute more information than the cp genomes, furthermore, the Next generation sequencing technique is often used for indicating several high through-put sequencing methods, actually only information from chloroplast genomes directly helped the authors realize their research
aims.
Therefore, it is not necessary to emphase "the Next generation sequencing" in the title of the article.
A more clear and proper title of the article might be
Solving phylogenetic incongruences in genus Polystachya of Orchidaceae Juss using chloroplast genome sequences
or
The use of chloroplast genome sequences solving phylogenetic incongruences in genus Polystachya of Orchidaceae Juss
or something like these.

2. If sequencing results were compared using different methods for confirmation when the cp genomes were sequenced using the method mentioned in the MS.

---

## Round 0.2 · accepted · Accept

I agree with the reviewer in that, the authors carefuly addressed all the issues raised in the the first revision. I believe this manuscript is now ready to be accepted for publication.

# Reviewer 1 ·

Basic reporting

Fulfilled

Experimental design

Fulfilled

Validity of the findings

Fulfilled

Additional comments

The authors properly addressed each of the comments raised in the previous revision by complementing they literature search and by explaining in much more detail several points not clearly explained in the previous manuscript text.